# Monitoring life expectancy levels during the COVID-19 pandemic: Example of the unequal impact of the first wave on Spanish regions

Sergi Trias-Llimós[1,2]*, Tim Riffe[3], Usama Bilal[4,5]

**1** Center for Demographic Studies, Bellaterra, Spain, **2** Department of Non-Communicable Disease Epidemiology, Faculty of Epidemiology and Population Health, London School of Hygiene & Tropical Medicine, London, United Kingdom, **3** Max Planck Institute for Demographic Research, Rostock, Germany, **4** Department of Epidemiology and Biostatistics, Dornsife School of Public Health, Drexel University, Philadelphia, Pennsylvania, United States of America, **5** Urban Health Collaborative, Dornsife School of Public Health, Drexel University, Philadelphia, Pennsylvania, United States of America

* strias@ced.uab.cat

## Abstract

### Background

To provide an interpretable summary of the impact on mortality of the COVID-19 pandemic we estimate weekly and annual life expectancies at birth in Spain and its regions.

### Methods

We used daily death count data from the Spanish Daily Mortality Monitoring System (MoMo), and death counts from 2018, and population on July 1st, 2019 by region (CCAA), age groups, and sex from the Spanish National Statistics Institute. We estimated weekly and annual (2019 and 2020*, the shifted annual calendar period up to 5 July 2020) life expectancies at birth as well as their differences with respect to 2019.

### Results

Weekly life expectancies at birth in Spain were lower in weeks 11–20, 2020 compared to the same weeks in 2019. This drop in weekly life expectancy was especially strong in weeks 13 and 14 (March 23rd to April 5th), with national declines ranging between 6.1 and 7.6 years and maximum regional weekly declines of up to 15 years in Madrid. Annual life expectancy differences between 2019 and 2020 also reflected an overall drop in annual life expectancy of 0.9 years for both men and women. These drops ranged between 0 years in several regions (e.g. Canary and Balearic Islands) to 2.8 years among men in Madrid.

### Conclusions

Life expectancy is an easy to interpret measure for understanding the heterogeneity of mortality patterns across Spanish regions. Weekly and annual life expectancy are sensitive and useful indicators for understanding disparities and communicating the gravity of the situation because differences are expressed in intuitive year units.

**Data Availability Statement:** All relevant data are within the manuscript, its Supporting information and and can be accessed here: https://osf.io/fwrk3/ .

**Funding:** STL acknowledges research funding from the HEALIN project led by Iñaki Permanyer (ERC-2019-COG agreement No 864616). UB was supported by the Office of the Director of the National Institutes of Health under award number DP5OD26429.

**Competing interests:** The authors have declared that no competing interest exist.

## Introduction

The COVID-19 pandemic is causing substantial increases in mortality in several populations worldwide. According to WHO, 2020 nearly 600,000 confirmed COVID-19 deaths occurred worldwide by July 19[th] 2020 [1]. Spain was one of the most affected countries with more than 28,000 deaths with laboratory confirmation of COVID-19 by that date [1]. Nonetheless, both the official number of COVID-19 cases and deaths, and the results from a recent seroprevalence study reveal important differences across Spanish regions [2, 3].

Beyond the official death toll statistics, the COVID-19 pandemic has been associated with net increases in mortality in several populations [4], which could owe to a combination of factors. The COVID-19 pandemic used an unprecedented amount of health service resources, including intensive care unit beds and strong preventive health measures in hospitals. This has put health systems in challenging situations [5], and thus potentially led to increases in morbidity and mortality indirectly related to COVID-19. For example, some excess mortality could result from healthcare avoidance, from treatment delays, or from insufficient care for other urgent conditions resulting from a reduced capacity to treat other medical emergencies. Other kinds of mortality may be temporarily reduced, such as deaths from acute respiratory conditions related to air pollution, or traffic accidents, but those mortality reductions may be outweighed by excesses. While total excess mortality and total number of COVID-19 deaths provide some measure of the impact of the pandemic within populations, their interpretation is not always straightforward, and comparisons between populations can be challenging.

Few studies assessing the impact of the pandemic on mortality to date have reported life expectancy estimates. Life expectancy is a summary index of mortality that is easy to interpret, but that is also subject to common misunderstandings. The most literal understanding of life expectancy is as the expected average length of life that would follow in the long run if a given set of mortality conditions were held fixed. The most common misunderstanding is to treat it as a forecasted expectancy. In monitoring mortality patterns in non-crisis times, annual life expectancy is a standard indicator because it accounts for differences in age-specific mortality, and it is expressed in intuitive year units, making it easy to grasp and compare across populations and over time.

The current pandemic, with fast mortality increases in specific weeks, makes it useful to shorten the calendar reference period of mortality rates in order to measure life expectancy changes. When calculated over short time intervals in this way, life expectancy should be understood as an annualized summary index of mortality. This index is much more volatile than standard period life expectancy calculated over a year, and it should be reported alongside traditional calendar-year life expectancy measures, possibly with shifting calendar reference windows. A few studies that have documented life expectancy declines have found notable declines in annual life expectancy in highly affected Spanish and Italian regions, Madrid and Bergamo [6, 7], in England and Wales [8], and declines in weekly life expectancy in Sweden [9].

The objective of this study is to estimate the impact of the first wave of the COVID-19 pandemic by estimating both weekly and annual life expectancies in Spain and its 17 regions.

## Methods

### Settings

We estimated life expectancy at birth by sex in Spain and its 17 regions (*comunidades autónomas*) in two time frames: i) weekly life expectancy at birth from week 1 in 2019 until the most recently available weekly data (week 27, June 28-July 5); and ii) annual life expectancy for both

2019 and the shifted annual reference period up to 5 July, 2020 (referred as 2020*). Due to small population sizes, we excluded the Spanish cities of Ceuta and Melilla on mainland Africa from the analyses.

## Data

Data sources included daily age- (<65, 65–75, 75+) and sex-specific death counts data from the Spanish *Sistema de vigilancia de la mortalidad diaria* (Daily Mortality Monitoring System, MoMo, updated July 16th) covering ~93% of the population [10]. We also used sex- and age-specific (5-year age groups) death counts by region in 2018 from Spanish National Statistics Institute (INE) [11], and population estimates from July 1st, 2019 from INE [12]. To compare changes in life expectancy with the number of infected people, we obtained data on the region-level seroprevalence of IgG against SARS-COV-2 from the seroprevalence study in Spain carried out in over 60,000 individuals between April 27 and May 11 2020 [3].

## Methods

We conducted our analysis in five steps. First, we grouped daily death counts into weeks. Second, we redistributed the MoMo death counts from broad age groups (<65, 65–75, and 75+) into 5-year age groups using 2018 death counts from the INE as a proportional standard. Third, we estimated age-specific death rates for each population group using the 2019 mid-year population as the denominator for annual estimates, and the population divided by (365/7) as denominator for weekly estimates. Fourth, both weekly and annual life expectancies were estimated using conventional life table techniques, and 95% confidence intervals were estimated based on the 2.5 and 97.5th percentiles of 1,000 random binomial deviates [13, 14]. Finally, we derived the expected variation in annual life expectancy between 2019 and 2020* by subtracting life expectancy at birth in 2019 from life expectancy at birth in 2020*.

To assess the robustness of our estimates we visually inspected the associations between the annual life expectancy variation and the IgG anti SARS-Cov2 prevalence by region and sex.

## Results

In Spain, weekly life expectancies at birth for weeks 11–20, 2020 were lower than those from the same weeks in 2019 (Fig 1, and **S1 Fig** in S1 Appendix). This drop in weekly life expectancy was especially strong in weeks 13 and 14 (23 March to 5 April), with national declines ranging between 6.1 and 7.6 years. This drop in weekly life expectancy was heterogeneous across Spanish regions. Madrid experienced the largest drop, which ranged between 11.2 and 14.8 years in weeks 13–14 for both men and women. Catalonia also experienced important drops, ranging between 8.4 and 9.4 years, along with Castile-La Mancha and Castile and Leon (**S2 Fig** in S1 Appendix). North-western regions of Galicia and Asturias, along with Murcia and the Canary Islands, did not experience major disruptions in weekly life expectancy. From weeks 21 (May 18–24) onwards, weekly life expectancy in 2020 has been at levels close to 2019 in Spain and in most of its regions, including Madrid and Catalonia.

Annual life expectancy differences between 2019 and 2020* also reflected an overall drop of 0.9 years (10.5 months) for both men and women (Fig 2, and **S3 Fig** in S1 Appendix). However, this difference was also heterogeneous across regions. Declines were steepest for Madrid, with a loss of 2.8 (95%CI: 2.6–2.9) years for men and 2.1 (2.0–2.3) years for women. The least affected regions (Canary and Balearic Islands, Andalusia, and Galicia) show no change in annual life expectancy in 2020* compared to 2019.

Visual inspection on the robustness of our results suggested the observed declines in annual life expectancy to be well aligned with the seroprevalence study in Spain carried out in over

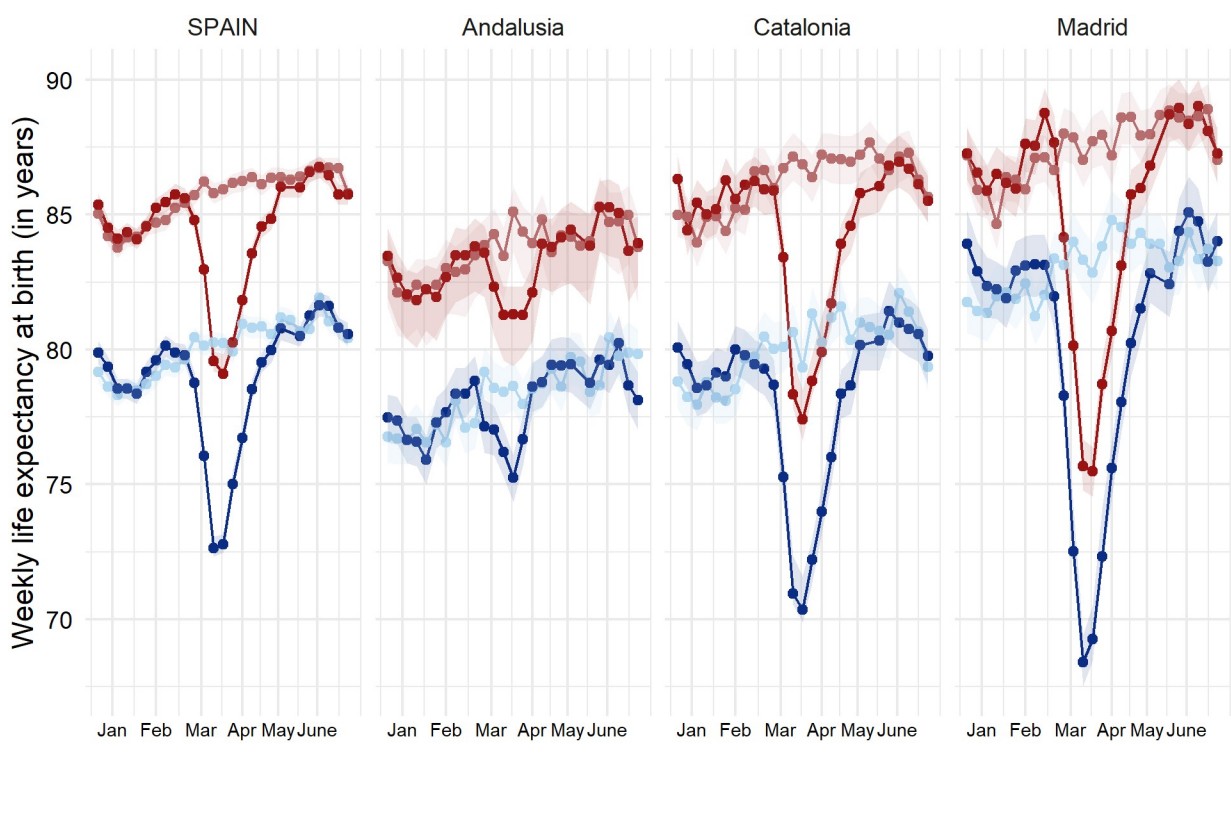

**Fig 1. Weekly life expectancy at birth (with 95% confidence intervals) in Spain and three selected regions (Andalusia, Catalonia and Madrid)[a] by sex (weeks 1–27, 2019 and 2020).** [a] Weekly life expectancies at birth for Spain and its 17 regions can be found in **S1 Fig** in S1 Appendix.

60,000 individuals between April 27 and May 11 2020 (2) (Fig 3). The Pearson correlation coefficient was 0.95 and 0.92 for men and women, respectively.

## Discussion

In this study, we have documented for the first time the impact of the COVID-19 pandemic on weekly and annual life expectancies at birth at the regional level in Spain, one of the most severely affected countries in the world as of mid-2020. The heterogeneity of the pandemic across Spain is reflected in wide differences between regions, with some of the most affected regions losing up to 10–15 years of weekly life expectancy and more than 2 years in annual life expectancy, whereas other regions' life expectancy was hardly affected. Overall, the observed drops in life expectancy are remarkable given that the vast majority of deaths with COVID-19 and excess mortality during the pandemic occurred at older adult ages [2, 10]. The observed changes in life expectancy correlated strongly with the prevalence of antibodies against SARS-COV-2.

Our estimates of a 0.9 year decline in annual life expectancy in Spain suggest that the COVID-19 pandemic has the potential to cause life expectancy stalls not seen for decades. The available evidence from similar studies for other populations suggests that, at the country level, this impact may be larger in England & Wales, nearly 2 years [8]. Furthermore, a recent simulation study suggested increased direct effects of COVID-19 on life expectancy worldwide for

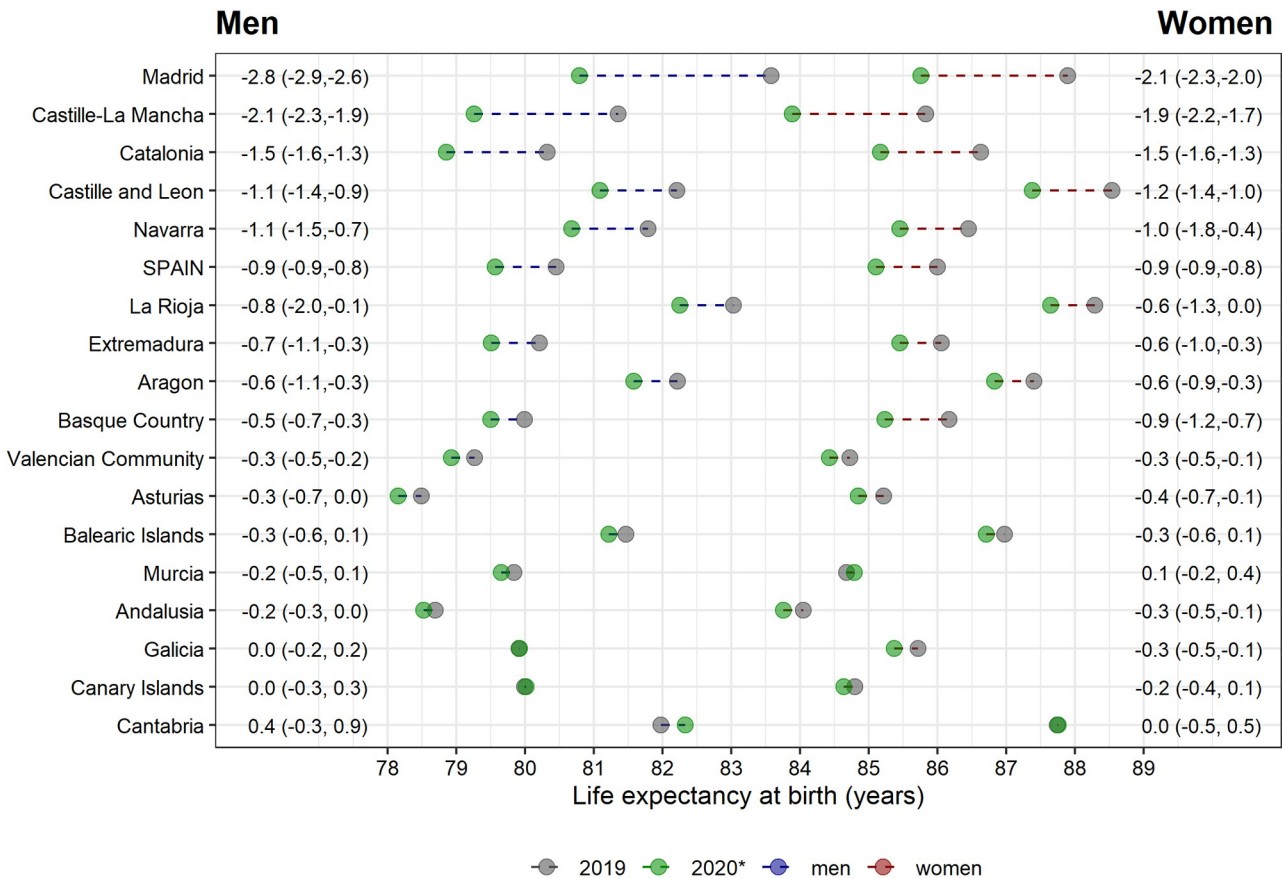

**Fig 2. Annual life expectancy at birth in 2019, 2020*[a] and differences between periods for Spain and its 17 regions by sex.** [a] Annual life expectancy at birth in 2020* was estimated using death counts from the shifted annual reference period up to 5 July 2020.

populations with COVID-19 prevalence around or higher than 5% [15]. Altogether, our work contributes to the growing body of literature suggesting COVID-19 to have the potential to strongly affect annual life expectancy.

Our estimates of the 2–3 years drop in annual life expectancy for Madrid, the most severely hit region, appear slightly lower than similar estimates from the Italian province of Bergamo up to April 30th, 2020 [7]. However, this comparison should be taken cautiously as the time frame and pandemic timing differed between studies, and life expectancy was compared to 2017 in the Bergamo study whereas we compared with 2019 levels. Our estimates of a 2–3 year drop in life expectancy in the most affected weeks in some of the least affected regions (e.g. Andalucia or Cantabria) (see **S4 Fig** in S1 Appendix) are comparable with a previous study for Sweden [9], in line with the higher impact of COVID-19 in Spain compared to Sweden [1].

There are plausible reasons to think that mortality in the remainder of 2020 may further increase compared to 2019. The epidemic is not yet over at the time of this writing and a second wave with a dramatic increase in newly detected cases is occurring in August and September 2020 [16]. Nonetheless, at the time of writing (mid-September 2020) the number of deaths with COVID-19 remain at low levels, albeit increasing. Furthermore, excess of mortality has not yet been observed [10]. This second wave could evolve differently than the first one in terms of mortality impact, age profiles, and geographical differences across the country. Last, the effects of delayed care of chronic conditions, cumulative anxiety, alcohol consumption,

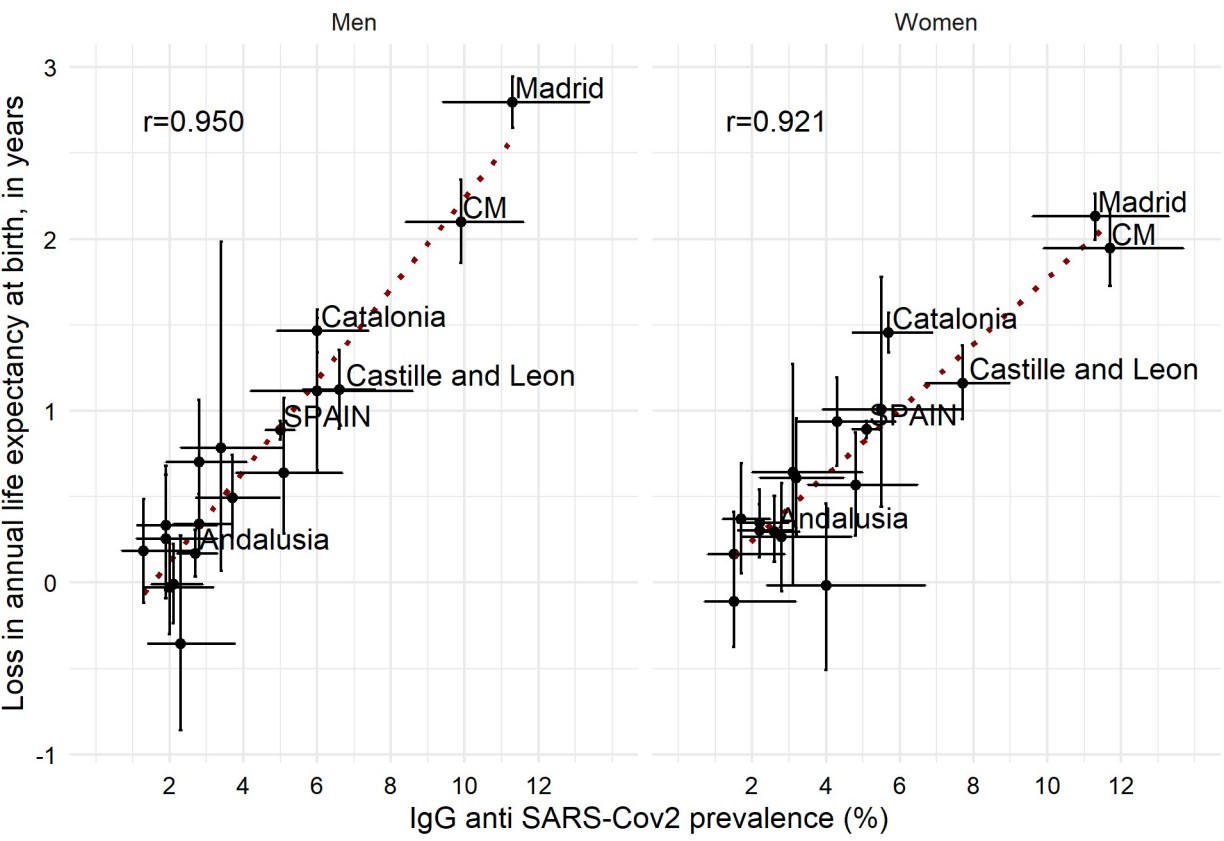

**Fig 3. Associations between loss in annual life expectancy at birth between 2019 and 2020[*][a] and IgG anti SARS-Cov2 prevalence for Spain and its 17 regions by sex (95% CI).** [a] Annual life expectancy at birth in 2020* was estimated using data from the one year window up to 5 July 2020. [b]CM stands for Castille-La Mancha.

and other factors during the pandemic may contribute to increased mortality in the coming months.

However, mortality may also decrease in the coming months due to the mortality selection of frail individuals which usually occurs after severe flu episodes. That is, the COVID-19 pandemic has been more fatal among elderly individuals with pre-existing health conditions [17, 18] as well as individuals residing in nursing homes [18, 19]. Therefore, it is plausible that some individuals who would have been expected to die in the remaining part of the year have already died, which could lead to mortality reductions in the second half of 2020. Indeed, this seems to occur in the regions of La Rioja and Navarra, where increases in weekly life expectancy are observed in the last week of May and first weeks of June as compared to 2019 (**S1 and S4 Figs** in S1 Appendix). We do not yet know to what extent such processes might offset one another, and careful monitoring of weekly life expectancy for the remainder of the year should provide answers on the overall impact of the COVID-19 pandemic.

In the first half of 2020, Spain has been one of the most affected countries both in terms of directly related COVID-19 deaths [1], as well as in terms of total excess mortality [4]. We estimated the annual life expectancy drop between 2019 and the one year window that closes out on 5 July, 2020 to be 0.9 years (~11 months) in Spain. In contrast, the average annual increase in life expectancy in Spain increased on average two months per year from 2009 to 2019. Altogether, this suggests that life expectancy drop between observed and expected annual life expectancy in the recent one year window would be around or below one year.

Other populations seem to be as affected or more affected by COVID-19 than Spain. For instance, the UK had the highest relative excess mortality [20], and important geographical inequalities exist as well, with London leading the negative ranking of relative excess mortality [21]. Indeed, metropolitan areas with important public transport networks tend to be more affected than other regions within a country, not only in Spain or the UK, but also in Italy, where Lombardy was by far the most affected region [22]. Other highly affected metropolitan areas include New York city, where the excess mortality was three-fold higher during seven weeks [23]. At the time of writing, other populations, for example Brazil, Ecuador, or Chile have also been substantially affected by COVID-19 [24]. If the age distribution of deaths in other populations skews younger, this may lead to stronger impacts on life expectancy in these populations. Monitoring weekly and annual life expectancies in the most affected populations would provide clear and understandable information on the impact of the pandemic on mortality at the population level. This information enables comparing impacts across populations.

Interestingly, our results suggest a similar mortality drop for men and women (0.9 years, in annual life expectancy). This finding contrasts with estimates from Italy, Sweden, and England and Wales, where the largest life expectancy drops–although of different magnitude- were observed for men compared to women [7–9]. Whereas a typical sex-gap was observed in some of the most affected regions (e.g. Madrid or Castille-La Mancha), it was not observed in some of the least affected regions (in Fig 2: Cantabria 0.4 years for men and 0.0 for women; Galicia 0.0 years for men and -0.3 years for women; or Andalusia -0.2 years for men and -0.3 years for women). Although this may be counterbalancing the national results, it is unlikely to entirely explain the observed similar life expectancy drop between men and women. Looking for explanations of this rather comparatively high mortality among women different factors may be at play. First, Spanish women have the highest life expectancy in Europe [25], but rank 9[th] in healthy life expectancy at age 65, indicating a higher rate of comorbidities and potentially a higher vulnerability to COVID-19 [26]. In fact, while life expectancy at birth in 2018 was 5.6 years higher in Spanish women as compared to men, healthy life expectancy at age 65 was 0.2 years lower in women compared to men. Second, nursing homes, mostly populated by women [27], were particularly affected by COVID-19 in Spain [19, 28]. Further potential explanations for this gap would require analysing cause-of-death and place-of-death mortality data, which is unfortunately not available for Spain at the time of writing.

Our analyses are based on detailed daily death counts data covering 93% of the population. This could result in slightly overestimated life expectancy levels, especially for the four regions with real coverage <80% (Aragon, Cantabria, Castile and Leon, and La Rioja). Therefore, life expectancies per se need to be interpreted cautiously. However, the undercoverage of the data used as well as the assumption of the age-specific patterns from 2018 are unlikely to substantially affect our main outcome, the differences between life expectancies (see **Appendix II** in S1 Appendix for details and for sensitivity analyses exploring the impacts of undercoverage). In our analyses we have assumed the age pattern of deaths in 2018 applies to the first half of 2020 (2020*). In a sensitivity analysis we used age-specific mortality estimates recently released by the INE, corrected by the undercoverage of electronically reported mortality data (~7%) [29], and we obtained similar estimates on the life expectancy gaps between 2019 and 2020* (Fig 4). Thus, these additional analyses confirm that both the undercoverage and our assumptions regarding the age pattern of deaths are unlikely to affect he estimated life expectancy differences between 2019 and 2020* (data from the one year window up to 5 July 2020).

The weekly life expectancy estimates presented in this study summarize the intensity of mortality increases [8]. Weekly life expectancy is a sensitive, intuitive, and comparable

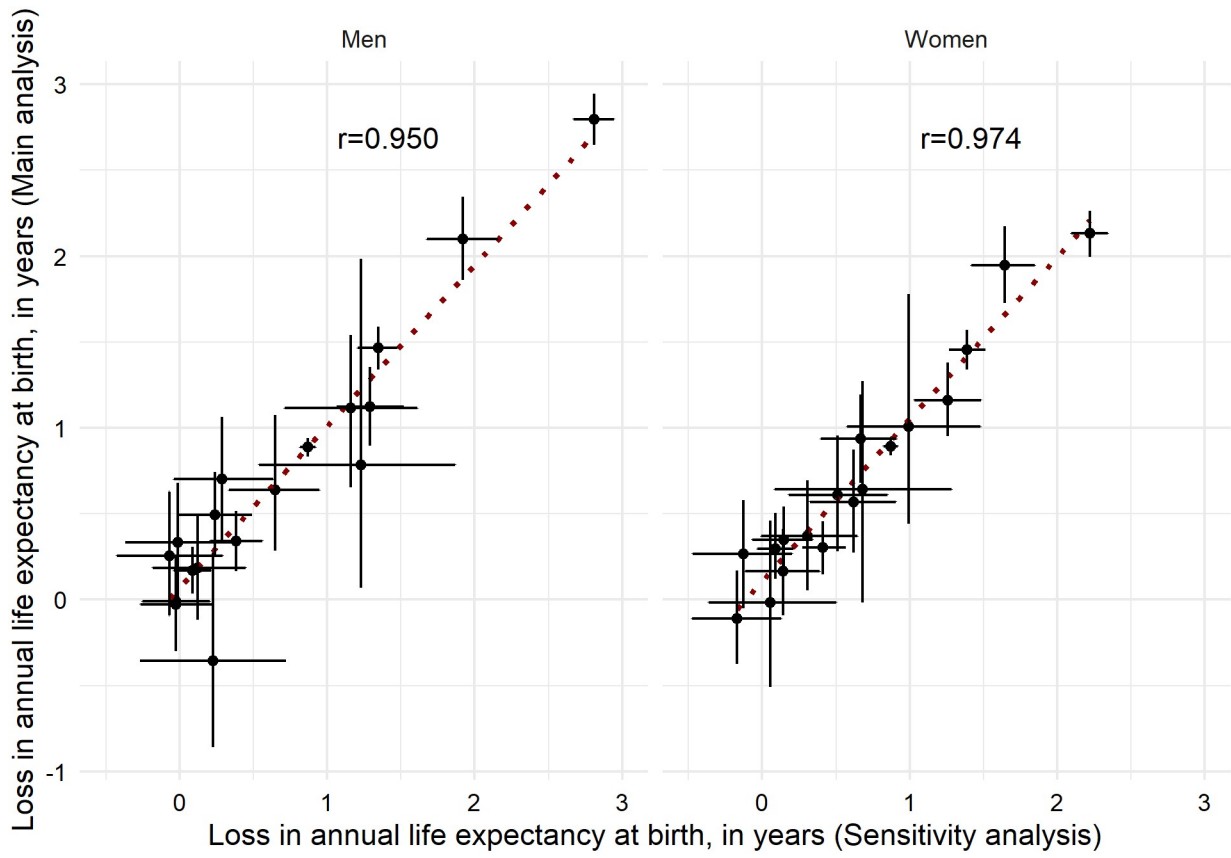

**Fig 4. Life expectancy at birth changes between 2019 and 2020\*[a]: Comparison between our main results and results derived from INE.** [a] Annual life expectancy at birth in 2020\* was estimated using data from the one year window up to 5 July 2020.

translation of mortality rates. Weekly-estimated life expectancies can be compared with standard annual life expectancies and with similar estimates from this or other mortality shocks, but one must be careful not to overinterpret this index as a forecast. For example, our results for Madrid on the weekly life expectancies in weeks 13 and 14 shows a substantial drop of up to 15 years, while the annual life expectancy for the one year window that closes out on 5 July, 2020 (week 27) shows a 2.8 year drop for men. This is not a provisional estimate of the 2020 life expectancy impact, which would require a forecast of mortality through the end of the calendar year.

In conclusion, the impact of COVID-19 pandemic has been severe and highly heterogeneous in Spain. Weekly and annual updated life expectancy are valuable indicators of the health impacts of the pandemic in populations, and thus should be continuously monitored. Such monitoring efforts should be sustained by up-to-date information on all-cause mortality disaggregated by age and sex. Detailed and updated mortality data should be released by public health agencies and governments worldwide [30, 31], requiring an increase of the coverage of electronic vital event reporting and seek to reduce other sources of reporting lags.

## Supporting information

**S1 Appendix.**
(DOCX)

## Author Contributions

**Conceptualization:** Sergi Trias-Llimós, Tim Riffe, Usama Bilal.

**Data curation:** Sergi Trias-Llimós.

**Formal analysis:** Sergi Trias-Llimós.

**Investigation:** Sergi Trias-Llimós, Tim Riffe, Usama Bilal.

**Methodology:** Sergi Trias-Llimós, Tim Riffe, Usama Bilal.

**Project administration:** Sergi Trias-Llimós.

**Software:** Sergi Trias-Llimós.

**Supervision:** Tim Riffe, Usama Bilal.

**Validation:** Sergi Trias-Llimós.

**Visualization:** Sergi Trias-Llimós, Usama Bilal.

**Writing – original draft:** Sergi Trias-Llimós.

**Writing – review & editing:** Tim Riffe, Usama Bilal.

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
