## [Decision Letter · Decision Letter 0]

1 Sep 2020

PONE-D-20-22554

Monitoring life expectancy levels during the COVID-19 pandemic: Example of the unequal impact in Spanish regions

PLOS ONE

Dear Dr. Trias-Llimós,

Thank you for submitting your manuscript to PLOS ONE. After careful consideration, we feel that it has merit but does not fully meet PLOS ONE’s publication criteria as it currently stands. Therefore, we invite you to submit a revised version of the manuscript that addresses the points raised during the review process.

We look forward to receiving your revised manuscript.

Kind regards,

Kannan Navaneetham, PhD

Academic Editor

PLOS ONE

Journal Requirements:

"STL acknowledges research funding from the HEALIN project led by Iñaki Permanyer (ERC- 2019-COG agreement No 864616). UB was was supported by the Office of the Director of the National Institutes of Health under award number DP5OD26429."

"No.

The funders had no role in study design, data collection and analysis, decision to

publish, or preparation of the manuscript."

Reviewers' comments:

Reviewer's Responses to Questions

**Comments to the Author**

1. Is the manuscript technically sound, and do the data support the conclusions?

Reviewer #1: Partly

Reviewer #2: Yes

2. Has the statistical analysis been performed appropriately and rigorously? 

Reviewer #1: Yes

Reviewer #2: Yes

3. Have the authors made all data underlying the findings in their manuscript fully available?

Reviewer #1: Yes

Reviewer #2: Yes

4. Is the manuscript presented in an intelligible fashion and written in standard English?

Reviewer #1: No

Reviewer #2: Yes

5. Review Comments to the Author

Reviewer #1: Review of the manuscript draft

“Monitoring life expectancy levels during the COVID-19 pandemic: Example of the unequal impact in Spanish regions”

by Sergi Trias-Llimós, Tim Riffe and Usama Billal

Overview

- Objectives. The authors evaluate the provisional human cost of the Covid-19 pandemic in Spanish regions by estimating and comparing weekly and annual life expectancies in 2020 vs 2019. Life expectancy allows them to provide a concise and immediate measure of the provisional death toll of the pandemic in Spain. By conducting the analysis at the sub-national level, they are able to investigate the spatially heterogeneous impact of Covid-19 across the national territory.

- Data and Methods. The authors use daily sex-specific death counts data, aggregated over broad age groups (<65, 65-75, 75+), provided by the Spanish MoMo, relative to years 2019 and 2020. They redistribute such deaths counts over 5-year age groups, applying proportions based on regional death counts relative to year 2018 provided by INE. They estimate region-, sex-, and age-specific mortality rates using the 2019 mid-year population as denominator for annual estimates, and the same population divided by 365/7 for weekly estimates. They use such rates to derive life expectancy estimates through conventional life table methods and compute the differences between 2020 and 2019 estimates.

- Results. The authors document substantial heterogeneity in the death toll of Covid-19 across Spanish regions. In Madrid, the most affected region, the sharpest drop in weekly life expectancy (11.2 to 14.8 years) is recorded in weeks 13-14, while the estimated drop in annual life expectancy is 2.8 years in the case of men and 2.1 years in the case of women. In contrast, least affected regions display no major disruption in weekly and annual life expectancy.

- Main contribution. The authors provide previously unavailable evidence on the provisional human cost of the Covid-19 pandemic in Spain, one of the most affected countries in Europe so far, and on its heterogeneity over the national territory.

- Main strengths

1. Use of overall death counts from all causes allows to capture the full impact of the Covid-19 pandemic, without suffering from issues associated to official Covid-19 deaths statistics (e.g. underreporting, exclusion of indirect mortality from other pathologies due to treatment delay, etc.).

2. Focus on regions allows to better uncover the mortality consequences of Covid-19. Indeed, given the spatial clustering of the pandemic, national figures tend to underestimate the impact on public health.

3. Likewise, use of daily death counts allow to derive weekly life expectancies which provide a timely measure of Covid-19 death toll.

- Main weaknesses

1. Use of daily death counts originally aggregated over broad age groups (<65, 65-75, 75+) prevents a finer analysis of contribution of various age-groups to overall mortality. In fact, the disaggregation into finer ager groups as described by the authors is somewhat problematic (see below).

2. Little discussion about the impact on men vs women and across age groups.

3. Some methodological aspects about computation of annual mortality rates are not clear, i.e. the reference period and the population used as denominator (see below).

Major comments

1. The disaggregation of daily death counts from broad age groups (<65, 65-75, 75+) into 5-year age groups presents some issues. By applying standard proportions derived from the 2018 regional death counts, the authors may introduce a bias in the derivation of age-specific mortality rates for 2020. Indeed, mortality due to Covid-19 is largely concentrated among people aged 50+ (at least so far). If most of the increase in 2020 deaths comes from middle-aged/elderly people, applying proportions based on 2018 death counts may result into an over-assignment of deaths to younger age groups, and consequently an over-estimation of the drop in life expectancy. The authors should mention and discuss this issue.

2. There is little discussion on the difference in mortality between men and women. Results from other countries suggest that mortality among men is disproportionately higher than among women, at all ages (see, for instance, Modig and Ebeling, 2020; Ghislandi et al., 2020). The authors do not discuss this point in the manuscript. In fact, they find that the estimated drop in (annual) life expectancy is remarkably similar for men and women across most Spanish regions (Figure 1). This point deserves further attention. In particular, the authors should discuss why the mortality pattern of Covid-19 in Spain is not as gendered as in other European countries.

3. It is not entirely clear which is the period over which the shifted life expectancy (2020*) is computed. In line 86, they write “[…] up to 5 July, 2020”. In line 216, they refer to the period “[…] May 2019 to May 2020 […]”. This point should be clarified, possibly in the Methods section.

4. It is not entirely clear which population measure is used as denominator in the computation of age-specific mortality rates for shifted annual estimates. The authors state that they use 2019 mid-year population for annual estimates. Do they use it also to compute annual life expectancy between May 2019 and May 2020, mentioned in line 216? If yes, does this potentially affect their estimates in a substantial way?

Minor comments

1. There is little discussion on the mortality impact of Covid-19 over different age groups. Since life expectancy is a summary of age-specific mortality rates, which weighs young-age deaths more than old-age deaths, the authors may stress that the estimated drops in life expectancy are particularly remarkable if one considers that deaths have (probably) occurred mostly at older ages.

2. Page 3, lines 43-44: a reference period should be added to the statement “Spain has been one of the most affected countries with more than 28,000 deaths with laboratory confirmation of COVID-19” (28,000 deaths to which date?).

3. Page 5, line 85: should be clarified that most available weekly data refer to current year (i.e. week 27 of 2020).

4. Page 9, lines 148-150: the meaning of the sentence “The little available evidence […], nearly 2 years” is not clear. Do the authors mean that the impact of Covid-19 on LE in Spain is larger than in England & Wales, or the contrary?

5. Page 12, line 216: the 2.8 year drop is for men? It should be specified.

6. Page 18, line 336: there is probably a typo in the sentence “that up to…”.

7. Lines 151 and 170: the authors employ the adjective “elevated”. I am not sure this adjective is used in English. Why not increased/high/higher?

8. Figure 1: why does the confidence bands for weekly LE in 2020 get much stricter (seem to disappear) for Catalonia and Madrid between March and April? This comment applies for the same regions also to Figure S4

9. Figure 2 and all Figures analogous to Figure 2: the legend is not straightforward. Maybe, the authors can use different symbols for changes in life expectancy (arrows) and point estimates (circles).

10. Figure 2: for the sake of clarity, the authors may specify which gender the columns reporting changes in LE and CIs refer to.

11. Figure S1: why there are not confidence bands?

12. Figure 2 in Appendix II: it could be useful to add confidence intervals to point estimates to show in which cases the two estimates are different in a statistically significant way.

Reviewer #2: The paper provides a very straightforward and relevant assessment on the impact of COVID-19 on life expectancy in Spain and its regions based on daily count of deaths. Overall, I have very positive feedback and would recommend publication after fixing some small issues.

1-In the calculation of the yearly life expectancy, what is assumed for the rest of the year (past week 20)? First, I thought you assume that the mortality is back to normal from week 20 until the end of the year, as you generally wrote “life expectancy in 2020”. When I read the last 2 paragraphs, I understood that the yearly life expectancy is indeed from May 2019 to May 2020 (thus no forecast) and not an estimate for 2020. On line 329, you then write that the period is until July 2020 (which would imply a small forecast). This is a bit confusing.

I would specify in the objectives that you assess the impact of the first wave of COVID-19 (from week 11 to 20) on life expectancy (otherwise, with the second wave that is rising, the paper might be quickly outdated). I also suggest being clearer in the method about the period used as reference in the yearly life expectancy. If you did some forecasts for the rest of the year, specify it more clearly.

2-Because things are evolving very quickly, I would be careful with statements such as “one of the most severely affected countries in the world”(142, 182 and many other places). In February, the most severely affected country was China. 5 months later, China was among the least affected countries per inhabitant. The situation can thus be radically different just a few weeks after the paper is published.

Similarly, authors should revise statements that are already outdated such as (196-198) “Other populations, for example Brazil, Ecuador or Chile are somewhat behind European countries in the pandemic at the time of this writing, but rapidly experiencing a dramatic death toll increase (24).”

Minor.

Lines 126, 131 and some other places: there are *, but it’s not clear what they stand for. I see on lines 324, 329 and 335 different meanings.

6. PLOS authors have the option to publish the peer review history of their article (what does this mean?). If published, this will include your full peer review and any attached files.

Reviewer #1: No

Reviewer #2: **Yes: **Guillaume Marois

---

## [Author Response · Author response to Decision Letter 0]

24 Sep 2020

Please, see the reply to the reviewers document for a point-by-point response to reviewers concerns

---

## [Decision Letter · Decision Letter 1]

26 Oct 2020

Monitoring life expectancy levels during the COVID-19 pandemic: Example of the unequal impact of the first wave on Spanish regions

PONE-D-20-22554R1

Dear Dr. Trias-Llimós,

We’re pleased to inform you that your manuscript has been judged scientifically suitable for publication and will be formally accepted for publication once it meets all outstanding technical requirements. Also  subject to minor amendments suggested by the reviewers before the production.

Kind regards,

Kannan Navaneetham, PhD

Academic Editor

PLOS ONE

Additional Editor Comments (optional):

Reviewers' comments:

Reviewer's Responses to Questions

**Comments to the Author**

1. If the authors have adequately addressed your comments raised in a previous round of review and you feel that this manuscript is now acceptable for publication, you may indicate that here to bypass the “Comments to the Author” section, enter your conflict of interest statement in the “Confidential to Editor” section, and submit your "Accept" recommendation.

Reviewer #1: (No Response)

Reviewer #2: All comments have been addressed

2. Is the manuscript technically sound, and do the data support the conclusions?

Reviewer #1: Yes

Reviewer #2: Yes

3. Has the statistical analysis been performed appropriately and rigorously? 

Reviewer #1: Yes

Reviewer #2: Yes

4. Have the authors made all data underlying the findings in their manuscript fully available?

Reviewer #1: Yes

Reviewer #2: Yes

5. Is the manuscript presented in an intelligible fashion and written in standard English?

Reviewer #1: Yes

Reviewer #2: Yes

6. Review Comments to the Author

Reviewer #1: The authors were able to effectively address the major issues pointed out in the first review. In particular:

• They were able to provide a convincing answer to the issues related to the use of age-specific mortality rates from 2018.

• They were able to provide some convincing explanations for why the mortality impact of the first wave of Covid-19 in Spain is not as gendered as in other European countries.

The authors can find comments relative to some minor issues below.

Minor comments

1. Disaggregation of daily death counts from broad age groups (<65, 65-75, 75+) into 5-year age groups. In the updated text, the authors implicitly acknowledge that using 2018 age-specific mortality rates could be problematic. For this reason, they show that there is a strong correlation (>0.9) between their estimates of LE drop and estimates of LE drop based on provisional age-specific mortality rates provided by INE. This evidence suggests that their estimates are, indeed, robust. Still, I think that for sake of research quality, the authors should explicitly state in the text why using age-specific mortality rates from 2018 could be problematic (i.e. that it may lead to over-estimation of LE drop).

2. Figure 2 and all Figures analogous to Figure 2: the legend is still not straightforward. In their reply to the reviewer, the authors say that they have removed the arrows, but in the updated version of the paper, there are still arrows in Figure 2, which seems identical to the one attached to the first version of the paper.

3. Figure 2: reiterating the comment from the first review round, the authors should specify which gender the columns with changes in LE and CIs refer to. In their reply to the reviewer, the authors say that they have labelled the columns, but in Figure 2 of the updated paper, columsn are not labelled.

Reviewer #2: The working paper cited in reference 15 is now published in a journal. Please update it https://journals.plos.org/plosone/article?id=10.1371/journal.pone.0238678

7. PLOS authors have the option to publish the peer review history of their article (what does this mean?). If published, this will include your full peer review and any attached files.

Reviewer #1: No

Reviewer #2: No

---

## [Editor Report · Acceptance letter]

28 Oct 2020

PONE-D-20-22554R1 

Monitoring life expectancy levels during the COVID-19 pandemic: Example of the unequal impact of the first wave on Spanish regions 

Dear Dr. Trias-Llimós:

I'm pleased to inform you that your manuscript has been deemed suitable for publication in PLOS ONE. Congratulations! Your manuscript is now with our production department. 

Kind regards, 

on behalf of

Professor Kannan Navaneetham 

Academic Editor

PLOS ONE